# Blockchain-Based Multiple Authorities Attribute-Based Encryption for EHR Access Control Scheme

**Xiaohui Yang**  **and Chenshuo Zhang** *

School of Cyber Security and Computer, Hebei University, Baoding 071000, China
* Correspondence: xiao6zhuo@163.com

**Abstract:** The Internet of Medical Things (IOMT) is critical in improving electronic device precision, dependability, and productivity. Researchers are driving the development of digital healthcare systems by connecting available medical resources and healthcare services. However, there are concerns about the security of sharing patients' electronic health records. In response to the prevailing problems such as difficulties in sharing medical records between different hospitals and patients' inability to grasp the usage of their medical records, we propose a patient-controlled and cloud-chain collaborative multi-authority attribute-based encryption for EHR sharing with verifiable outsourcing decryption and hiding access policies (VO-PH-MAABE). This scheme uses blockchain to store the validation parameters by utilizing its immutable, which data users use to verify the correctness of third-party outsourcing decryption results. In addition, we use policy-hiding technology to protect data privacy so that data security is guaranteed. Moreover, we use blockchain technology to establish trust among multiple authorities and utilize Shamir secret sharing and smart contracts to compute keys or tokens for attributes managed across multiple administrative domains, which avoids a single point of failure and reduces communication and computation overhead on the data user side. Finally, the ciphertext indistinguishability security under the chosen plaintext attack is demonstrated under the stochastic prediction model and compared with other schemes in terms of functionality, communication overhead, and computation overhead. The experimental results show the effectiveness of this scheme.

**Keywords:** IOMT; blockchain; multi-authority attribute encryption; medical data sharing; policy hiding; outsourced decryption; outsourced verification



## 1. Introduction

A patient's electronic health record (EHR) is a personal medical health record that stores patient diagnostic and treatment information, such as medical records, allergy medications, physical examination reports, family medical history, and other sensitive information. When a patient visits a doctor, he or she can see the previous medical history and synthesize the previous diagnosis or treatment results to make a more comprehensive and accurate analysis of the condition and provide a more efficient treatment plan for the patient. Meanwhile, for major infectious diseases, the sharing of EHR can also enable excellent teams from various regions to conduct an all-round, accurate and rapid study of the epidemic situation, improving the efficiency of disposal and public medical health. However, the existing IOMT system can hardly satisfy the massive EHR data sharing security.

To satisfy the access control requirements of medical data, we propose ciphertext-policy attribute-based encryption (CP-ABE) and key-policy attribute-based encryption(KP-ABE) approaches to protect the security of shared data by encrypting the shared data using a key, and only the user with the key can decrypt the data. CP-ABE can better solve the interoperability challenges among the participants. Since data owners can use CP-ABE to flexibly specify access policies to determine which attributes are required for users to

decrypt, CP-ABE is considered an ideal solution for securely sharing outsourced data in public clouds. It is more suitable for access control scenarios for healthcare data.

However, in most existing CP-ABE programs [1–3], all participants trust an authority. Due to the centralized authority problem, the mischievous attribute authority (AA) even causes the misuse of private keys. It distributes the attribute keys to illegal users, leading to unauthorized data access. However, some multi-AA extension schemes did emerge later, dividing the attribute Universe into multiple administrative domains. Each AA manages a disjoint set of attributes, which gives rise to trust issues between attribute domains and does not fundamentally solve the single point of failure, as well as the data users bearing high computation and communication overheads. Moreover, in conventional CP-ABE, the access policy embedded in the ciphertext is publicly accessible, so an attacker can indirectly deduce sensitive identity information about the data owner and the data user [2]. In addition, CP-ABE schemes contain a large amount of bilinear computation during encryption and decryption, which consumes many computational resources and limits the application of CP-ABE [4].

The emerging blockchain technology brings a glimmer of light to these challenges. Blockchain is a distributed ledger technology with tamper-proof features, which enables anyone to host a distributed ledger and keep a permanent record of transactions. Moreover, blockchain technology can establish multi-party trust. Smart contracts deployed on the blockchain enable a collaborative computing process with multiple authorities from different attribute domains and generate attribute keys for users to decrypt data. Since there are performance bottlenecks in blockchain at this stage, and EHR usually includes large-scale, cross-media health data, such as CT, X-ray, and other medical image data, it is inefficient to store and share EHRs independently using blockchain. Thus, it is necessary to combine cloud storage and blockchain to complement each other for secure and efficient EHR sharing.

To address the above issues, we propose an EHR access control scheme (VO-PH-MAABE) for multi-authority attribute-based encryption with verifiable outsourced decryption and hidden access policies, which works as follows.

(1) We propose an outsourced decryption method in which the user uses the verification parameters stored in the blockchain to quickly verify the third-party outsourcing results, ensuring the correctness of the outsourced decryption results, and reducing the computational cost at the user's end;

(2) We hide the access policy to effectively prevent the user's specific attribute values from being leaked to third parties and ensure user privacy security;

(3) We use blockchain technology to build trust among multiple authorities and four smart contracts to compute keys or tokens for attributes managed across multiple administrative domains, which avoids a single point of failure and reduces the communication and computation overhead on the data user side.

## 2. Related Work

### 2.1. Multi-Authority Attribute-Based Encryption

To solve the problem of a single authority and distributed management of attributes. Lewko and Waters [5] proposed a CA-free stepwise multi-authority design that uses CP-ABE with no communication among authorities and operates independently. Li et al. [6] implemented a multi-AA CP-ABE scheme TMACS for public cloud storage using (t, n) threshold secret sharing. Zhong et al. [7] proposed a decentralized multi-authority CP-ABE access control scheme. Each AA distributes attributes independently in its management domain and uses an obfuscated access matrix to protect access policy privacy. Li et al. [8] proposed a privacy-aware multi-AA CP-ABE scheme with recourse to trace the identity of dishonest users who share their decryption keys by hiding attribute information in the ciphertext. Zhang et al. [9] introduced MA-ABE into a smart grid environment, which adds a test phase to determine whether a client's attributes conform to access rules. Li et al. [10] proposed a design for addressing user key misuse and fine-grained access

supervision of encrypted IoT data on the cloud, which can resist selective plaintext attacks and overcome clients. Gao et al. [11] proposed a time-sensitive multi-authority attribute encryption scheme based on time sensitivity, where data users cannot decrypt the ciphertext before a specific time. The scheme is suitable for data sharing in time-sensitive access control scenarios.

### 2.2. Hiding Access Policy

In response to earlier CP-ABEs that only partially hide attribute values in the access policy but do not protect attribute names, Zhang et al. [12] designed a policy-hiding CP-ABE solution that performs attribute matching operations before full decryption. The data user uses a unique ciphertext component to test whether the attribute list satisfies the hidden access policy. The scheme has selective plaintext security under the DBDH and D-Line assumptions. Huang et al. [13] designed a CP-ABE scheme for a hiding policy. The CP-ABE scheme can compress the ciphertext length to a constant. Xiong et al. [14] designed a policy-hidden broadcast encryption scheme that supports the LSSS access structure and improves the performance of the policy. Zhang et al. [15] used CP-AB with a hidden access policy to improve the expressiveness of the policy. The scheme supports large-scale attribute collections, and the size and decryption cost of the public parameters in the scheme is constant. Liu et al. [16] used a linear secret sharing scheme to hide part of the policy. They use multiple privileges to resist the complicity attack caused by joint communication among users and protect the privacy of data in the IoT environment. Zeng et al. [17] proposed an effective ABE scheme for partial policy hiding and supporting a large attribute universe in the context of the internet of medical things (IoMT) ecosystem. The scheme shows only non-sensitive attributes and hides sensitive attributes, and the scheme can effectively track any user with the public decryption key. Zhang et al. [18] proposed a partial policy hiding scheme supporting key revocation and designed an algorithm to check whether the user attributes to match the access policy.

### 2.3. Outsourcing Decryption

In existing ABE schemes, the ciphertext size and decryption cost increase with the complexity of the attribute policy. Therefore, Green et al. [19] first proposed an outsourcing ABE scheme in 2011. They outsourced the complex ABE decryption computation to a third-party cloud server for decryption, which reduces the computation of the decryption process on the user side. Since the third-party cloud server is not fully trusted, the correctness of the outsourced decryption result needs to be verified. Lai et al. [20] proposed an ABE scheme that supports the verification of outsourced decryption by introducing a verification element in the ciphertext. The re-encryption key is generated based on the user key and sent to a third-party cloud server for decryption. Qin et al. [21] proposed a method to convert any ABE scheme with outsourced decryption into an ABE scheme with verifiable outsourced decryption with verifiability. Li et al. [8] proposed an accountable and verifiable outsourced decryption CP-ABE scheme. This scheme achieves accountability for the user key leakage problem and the verifiable outsourced scheme by transferring the key. Fan et al. [22] proposed a CP-ABE scheme for outsourced decryption in the IoT environment. Fog nodes assist in implementing verifiable outsourced decryption, which reduces the complexity of user computation. Guo [23] proposed a lightweight verifiable CP-ABE scheme applied to wireless body-sensing networks, which provides users with correctness verification of outsourced decryption. Liu et al. [24] proposed a blockchain-based searchable attribute-based encryption scheme BC-SABE that supports user outsourcing decryption functions. The blockchain system replaces the traditional centralized server, responsible for generating threshold parameters and key management. Zhao et al. [25] constructed an eHealth fine-grained access control system AC-FEH in a fog computing scenario, which uses fog nodes for data encryption and decryption operations to minimize the computational cost for data owners and users. Guo et al. [26] proposed a scheme with fine-grained access control, outsourced decryption, and ciphertext verification

with the help of cloud servers and blockchain in the IoMT ecosystem. The scheme is based on the chameleon hash function to construct data user private keys with conflict resistance, semantic security, and keyless exposure.

## 3. Preliminaries

### 3.1. Bilinear Maps

Let $q$ be a large prime, $G_1$ and $G_2$ be two multiplicative cyclic groups of order $p$ multiplicative cyclic groups, and $g$ be a generating element on $G_1$. A bilinear mapping [27] $e : G_1 \times G_1 \to G_2$ satisfies the following properties.

(1) Bilinear: $\forall a, b \in Z_p^*, u, v \in G1$, we have $e(g^a, g^b) = e(g,g)^{ab}$.
(2) Non-degeneracy: $\forall u, v \in G_1, e(u,v) \neq 1$.
(3) Computability: $\forall u, v \in G_1$, there exists an efficient algorithm to compute $e(u,v)$.

### 3.2. Determined Bilinear Diffie-Hellman Problem

The deterministic bilinear Diffie–Hellman (DBDH) problem [28] : $G_1$ and $G_2$ are two multiplicative cyclic groups of order $p$ and satisfy the bilinear mapping $e : G_1 \times G_1 \to G_2$, and $g$ is the generator of $G_1$. Given random numbers $a, b, c, z \in Z_p^*$ and group elements $g^a, g^b, g^c \in G_1, Z \in G_2$, we suppose no algorithm can generate a random number $a, b, c, z$ with non-negligible distinguish $Z = e(g,g)^{abc}$ in polynomial time with negligible advantage or $Z = e(g,g)^z$. In that case, the DBDH problem is considered intractable.

### 3.3. Access Structure

Let $U$ be an attribute Universe. A set $A \subseteq 2^U$ is said to be monotone provided that it satisfies the set $X \in A$ and the set $Y \subseteq 2^U$ when $X \subseteq Y$ and $Y \in A$. Similarly, a geometric element in an access structure with monotonicity forms an authorized set, and an element of a set not in the access structure forms a non-authorized set. A secret sharing scheme $\Pi$ over an attribute Universe is called a linear secret sharing scheme when it satisfies the following conditions:

(1) The segmentation of the secret value $s$ forms a vector. These partitions are all elements in $Z_p^*$. For $\Pi$, a shared generating matrix $M$ exists with $l$ rows and n columns. Function $\rho$ maps $1, 2, \ldots, l$ mapped to the attribute Universe, given that the vector $v = (s, r_1, \ldots, r_n), s \in Z_p^*$ is the secret value to be shared, where $r_2, \ldots, r_n$ are random elements in $Z_p^*$. $Mv$ is the $l$ secret value partitions over the secret value $s$. The $x$th secret value segmentation is noted as $\lambda_x$, which corresponds to the property $\rho(x)$.
(2) In order to reconstruct the secret value $s$, the user whose attributes satisfy the access policy can find a set of constants $\{\omega_1, \omega_2, \ldots, \omega_l\}$ in polynomial time such that $\sum_{x \in X} \omega_x M_x = (1, 0, \ldots, 0)$, where $X$ represents the set of rows corresponding to the user's attribute set $S$ in the matrix. The secret value is finally reconstructed according to the following equation.

$$\begin{aligned}
\sum_{x \in X} \omega_x \lambda_x &= \sum_{x \in X} \omega_x (Mv)_x \\
&= \sum_{x \in X} \omega_x (M_x)v \\
&= (1, 0, \ldots, 0)(s, r_2, \ldots, r_n) \\
&= s
\end{aligned}$$

### 3.4. Shamir Secret Sharing

Shamir secret sharing [29] requires the following:

(1) *Share.* To share a secret $s \in Z_p^*$ with a threshold $t$ among $n$ participants, we first construct a polynomial of order $(t-1) : f(x) = a_0 + a_1 x + \cdots + a_{t-1} x^{t-1}$, where $a_0 = s, a_1, \ldots, a_{t-1}$ are the random elements in $Z_p^*$. Then, the sharing of $n$ participants is $s_i = f(x_i)(i = 1, \ldots, n)$, where $x_i \in Z_p^*$.

(2) *Reconstruct.* In order to obtain the results from the *n* partitions $s_1, s_2, \ldots, s_n$ in any $t$ reconstructed secret values $s$, computed using Lagrangian interpolation:

$$S = \sum_{i=1}^{t=1} S_i \prod_{j=1, j \neq i}^{t} \frac{x_j}{x_j - x_i}$$

### 3.5. Smart Contract

As protocols for executing computer transaction contracts, smart contracts were proposed by Szabo [30] in 1994. It will always exist when a smart contract is integrated into a blockchain. Any smart contract can be a part of a database with a unique address. Distributing transactions to its address can enable its functionality to manage each part of the database. A smart contract is a set of software codes that the deployer specifies the conditions it is intended to implement. Smart contracts are often organized as "if . . . sthen. . . ". Smart contracts allow code to execute autonomously, without human intervention and third-party observation, when conditions are met. The contract deployer sets it up in the blockchain, and then the user enables it by sending the required parameters to the smart contract's address. In this scenario, the smart contract is invoked by a registered transaction in the blockchain.

## 4. Scheme Model

This section focuses on the system model, syntax, and security model of the VO-PH-MAABE scheme.

### 4.1. System Model

As shown in Figure 1, the VO-PH-MAABE system model includes six entities: certificate authorities (CAs), multi-attribute authorities (AAs), cloud servers (CSPs), data owners (DOs), data users (DUs), and Blockchain (BC).

Attribute authorities (AAs): AAs are responsible for publishing attributes to data users through the blockchain. There is a many-to-many mapping relationship between attributes and attribute management authorities. Each AA manages multiple attributes in an attribute domain, and multiple AAs can manage each attribute across domains. A segmentation of each secret is calculated and sent to the blockchain through secret sharing among AAs.

Cloud service provider (CSP): CSP provides data storage service for data owners (DO) and pre-decryption service for data users (DU) of cipher text. Because of its honest and curious nature, CSP is not trusted.

Data owner (DO): DO, also known as the patients themselves being responsible for encrypting data. They define the attribute-based access policy, encrypt the symmetric key by the access policy, and upload the resultant verification hash of the outsourced decryption to the blockchain. Then DO stores the TxID, data cipher, and key cipher returned by the blockchain to the cloud server.

Data user (DU): Data user wants to view patient cases, such as doctors, medical researchers, and insurance company managers. DU will register their attributes with AAs. In order to access the DO's data, the DU can initiate a pre-decryption request to the CSP. The decryption will be successful only if the user's attributes satisfy the access policy embedded in the cipher text. DU performs a power operation and hash operation on the pre-decryption result to verify the correctness of the outsourced decryption result.

Blockchain (BC): Blockchain stores public parameters and hashes for outsourced validation. After a user uploads a verification hash, the blockchain returns a transaction identifier TxID to the user. Blockchain also helps entities perform partially trusted computations and allows multiple AAs to collaboratively manage user attributes, which enables distributed management of attributes.

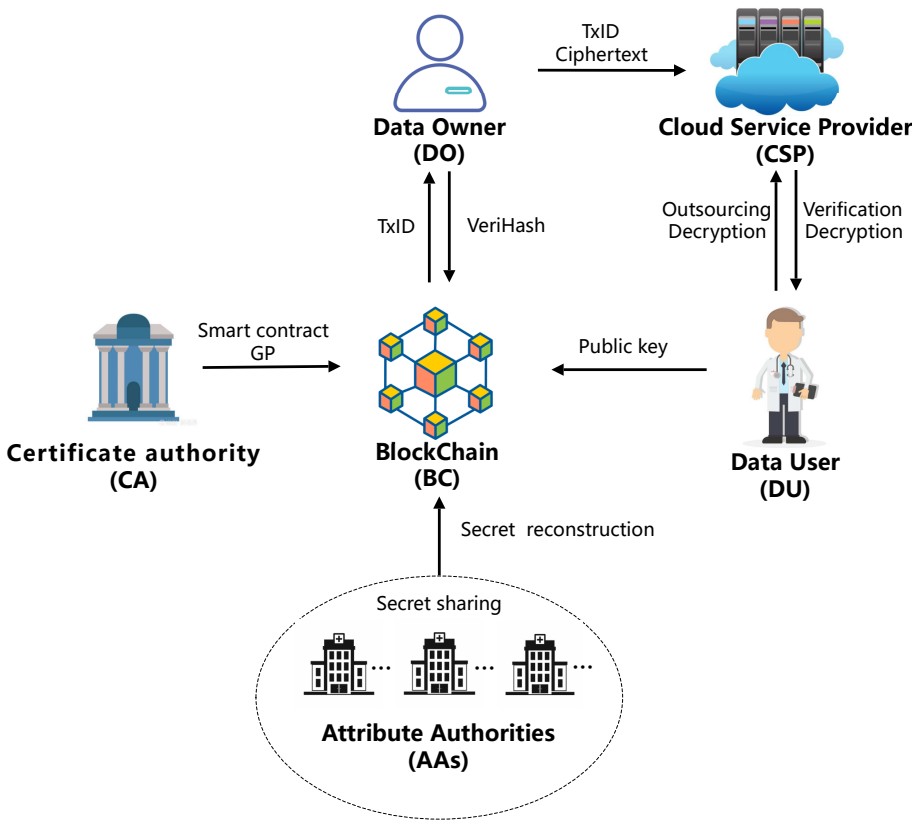

**Figure 1.** The VO-PH-MAABE system model.

*4.2. Syntax*

This VO-PH-MAABE scheme includes the following nine algorithms.

$GlobalSetup(1^{\lambda}) \rightarrow GP$: CA executes the global setup algorithm. This algorithm takes as input a security parameter $\lambda$ and outputs the system public parameter $GP$.

$AASetup(GP) \rightarrow (pk_i, sk_i)$: Different AAs execute the attribute authority setup algorithm using the Shamir secret sharing scheme. This algorithm takes as input the public parameters $GP$, then outputs the public and private keys$(pk_i, sk_i)$ for each $AA_i$.

$PAKGen(pk_i) \rightarrow PAK$: The on-chain contract executes the public attribute key generation algorithm. This algorithm takes as input the public key $pk_i$ from $AA_i$, then outputs the public attribute key $PAK$.

$DUSetup(GP) \rightarrow (upk, usk)$: DU executes the data user setup algorithm. This algorithm takes as input the public parameter $GP$, then outputs public and private keys$(upk, usk)$ for DU.

$UAKGen(GP, S_{uid}) \rightarrow UAK$: AAs and the on-chain contract jointly execute the user attribute key generation algorithm. This algorithm takes as input the public parameter $GP$ and an attribute set $S_{uid}$ for the data user with identity $uid$, then outputs the user attribute key.

$UAKGen(GP, S_{uid}) \rightarrow UAK$: The contract jointly executes the user attribute confused key generation algorithm. This algorithm takes as input the public parameter $GP$ and an attribute set $S_{uid}$ for the data user with identity $uid$, then outputs the user attribute confused key.

$Encryption(KEY, M_{data}, AP, GP, PAK) \rightarrow CT$: DO executes the encryption algorithm. This algorithm takes as input the symmetric key $KEY$, plaintext $M_{data}$, access policy $AP$, public parameters $GP$, and public attribute key $PAK$, then outputs ciphertext $CT$.

$OutDecryption(CT, SK_O, AP) \rightarrow CT'$: CSP executes the outsourcing decryption algorithm. This algorithm takes as input the ciphertext $CT$, access policy $AP$, and outsourcing decryption key $SK_O$, then outputs the outsourcing decryption ciphertext $CT'$. $SK_O$ is the user attribute key $UAK$ that satisfies the access policy $AP$.

*FullDecryption*$(CT', usk) \rightarrow M_{data}$: DU executes the full decryption algorithm. This algorithm takes as input the outsourcing decryption ciphertext $CT'$ and user private key *usk*, then outputs the plaintext $M_{data}$.

### 4.3. Security Model

The challenger generates a DBDH challenge. $\mathfrak{B}$ is a polynomial-time adversary attacking the DBDH problem, $\mathfrak{A}$ is a polynomial-time adversary attacking this scheme, $\mathfrak{B}$ uses $\mathfrak{A}$ to attack the DBDH problem, and $\mathfrak{B}$ is viewed as $\mathfrak{A}$'s challenger. We define the security concepts to show the security of the VO-PH-MAABE scheme. This scheme uses the concept of ciphertext indistinguishable security under adaptive selection plaintext attack (INDS-CPA) in the constructed proof. Including adversary $\mathfrak{A}$ and challenger $\mathfrak{B}$, the specific game is as follows.

*Initialization*: Adversary $\mathfrak{A}$ submits a challenge access policy $(M^*, \rho^*)$ to Challenger $\mathfrak{B}$.

*Setup*: Challenger $\mathfrak{B}$ chooses a sufficiently safe parameter $\lambda$ to run the $Setup(1^\lambda)$ algorithm to generate the public parameter $GP$. Challenger $\mathfrak{B}$ sends $GP$ to adversary $\mathfrak{A}$.

*Query Phase* **1**: The adversary adaptively queries the following Oracles. $O_{AAs}$: Adversary $\mathfrak{A}$ specifies a set of corrupted permissions, and for each attribute $\omega$ belonging to the uncorrupted permissions, $\mathfrak{B}$ runs $AASetup(GP, \omega)$ to generate the public attribute key $(e(g,g)^{\alpha_\omega}, g^{\beta_\omega})$ and returns it to $\mathfrak{A}$.
$O_{UAK}$: Adversary $\mathfrak{A}$ submits an identity and its corresponding attribute set $(uid, S^*)$ to $\mathfrak{B}$. When the attribute set is insufficient for the challenge access strategy, adversary $\mathfrak{B}$ runs $UAKGen(uid, (M^*, \rho^*), S^*, GP)$ to generate the attribute key $UAK_{\omega, uid}$, and returns it to $\mathfrak{A}$.

*Challenge*: $\mathfrak{A}$ submits two equal-length symmetric keys $KEY_0$ or $KEY_1$ to $\mathfrak{B}$. $\mathfrak{B}$ rolls a coin to determine the value of $\gamma \in \{0,1\}$ and executes $Encrypt(KEY_\gamma, M_{data}, (M^*, \rho^*), GP, e(g,g)^{\alpha_\omega}, g^{\beta_\omega}) \rightarrow CT^*$ and sends it to $\mathfrak{A}$.

*Query Phase* **2**: Adversary $\mathfrak{A}$ receives the challenge ciphertext $CT^*$ and continues to follow *Queryphase*1 adaptive querying.

*Guess*: Adversary $\mathfrak{A}$ guesses $\gamma' \in \{0,1\}$ for $\gamma$. If $\gamma = \gamma'$ then it is concluded that adversary $\mathfrak{A}$ wins. The advantage of adversary $\mathfrak{A}$ in winning this game is defined as $Adv_{VO-PH-MAABE}^{INDS-CPA}(\mathfrak{A}) = |Pr[\gamma = \gamma'] - 1/2|$.

If no adversary can break the above INDS-CPA game by a non-negligible margin in polynomial time, then the VOPH-MAABE scheme is considered INDS-CPA safe.

## 5. Scheme Implementation

This section describes in detail the system's interaction flow and algorithm design. Four types of smart contracts are introduced in this scheme: attribute management contract (AMC), public attribute key generation contract (PAKGC), user attribute key generation contract (UAKGC), and user attribute confused token generation contract (UACTGC). As shown in Figure 2, the overall flow of the implementation of this scheme is divided into four phases: system initialization phase, decryption key generation phase, encryption phase, and decryption phase.

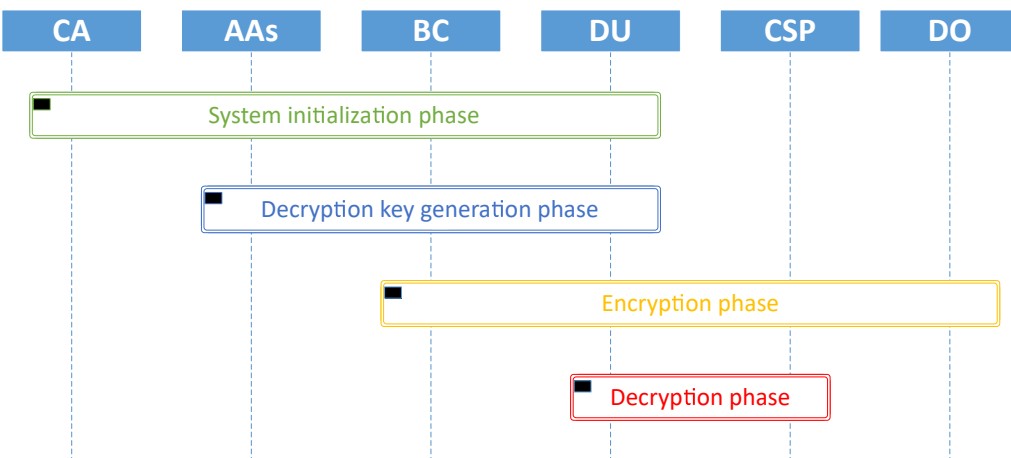

**Figure 2.** The overall flow of VO-PH-MAABE.

## 5.1. System Initialization Phase

As shown in Figure 3, the CA is responsible for global setup, and the AMC is responsible for managing attributes. AAs calculate the public attribute key segmentation off-chain, and PAKGC reconstructs the public attribute keys on-chain. DU is responsible for the data user setup.

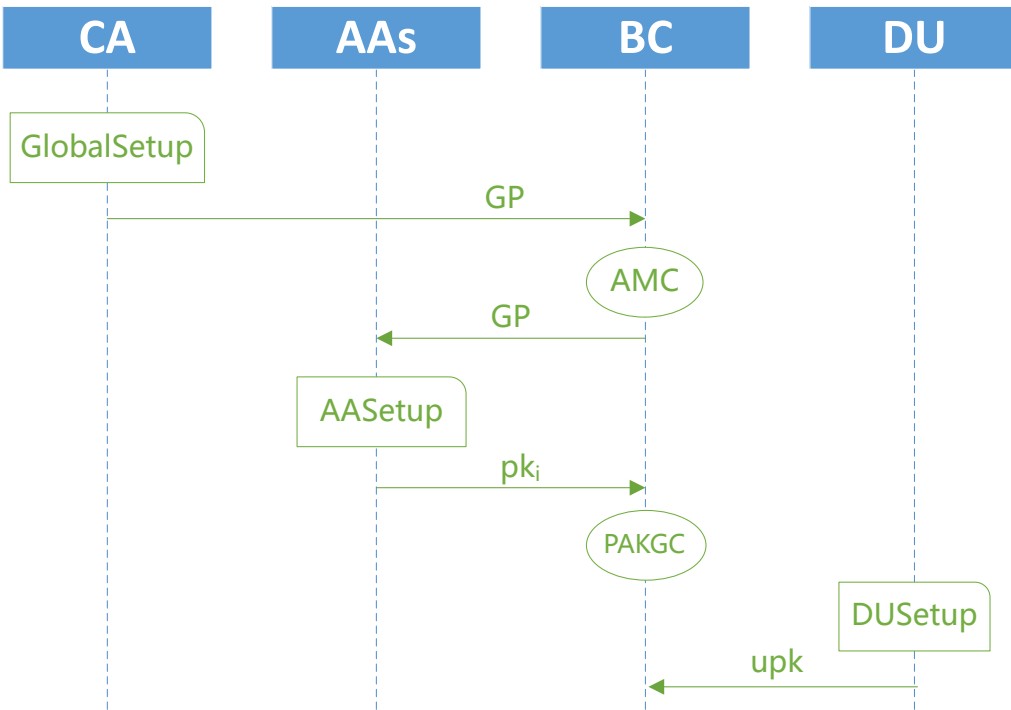

**Figure 3.** System initialization phase.

*Attribute Management Contract (AMC).* To enable cross-domain attributes management, CA assigns multiple AAs and threshold $t_\omega$ to each attribute by invoking the AMC. As shown in Algorithm 1, each attribute has two parameters: the number of AAs $n_\omega$ and the threshold $t_\omega$ for managing attribute $\omega$. The data user must obtain $t_\omega$ authorizations from the $n_\omega$ attribute authorities to access the attribute.

---

**Algorithm 1** Attribute management

---

1: //CA invokes the contract to assign a set of authorities and thresholds to attributes
2: **if** *Assignment*($\omega$, *AAs*, *t*) = true **then**
3:     $n_\omega \leftarrow$ a set of authorities;
4:     $t_\omega \leftarrow$ threshold;
5: **end if**

---

*GlobalSetup*($1^\lambda$) $\rightarrow$ *GP*: CA Select two prime p-order bilinear groups $G_1, G_2$, and $g$ is the generating element of G1. $e : G_1 \times G_1 \rightarrow G_2$ is a bilinear map. Then CA defines one-way hash function functions $H : \{0,1\}^* \rightarrow G_1$, $H' : G_2 \rightarrow Z_p^*$ and chooses random numbers $y \in Z_p^*$ to count $g^y$. The global parameters $GP = \{G_1, G_2, e, g, H, H', g^y\}$.

*AASetup*(*GP*) $\rightarrow$ (*pk_i*, *sk_i*): The AAs collaborate using the Shamir secret sharing scheme to obtain a key partition $\{pk_{i,\omega}, sk_{i,\omega}\}_{\omega \in S_i}$ for each attribute jointly managed using a list of AAs, where $S_i$ is the set of attributes managed by $AA_i$. For an attribute $\omega$ managed by $n_\omega$ AAs, $AA_i$ randomly chooses $\alpha_i, \beta_i \in Z_p^*$ to generate two $(t_\omega - 1)$ order random polynomials $F_i(x)$ and $H_i(x)$, and the master secret is defined as

$$s_1 = \alpha_\omega = \sum_{i=1}^{n_\omega} \alpha_i, \quad s_2 = \beta_\omega = \sum_{i=1}^{n_\omega} \alpha_i \tag{1}$$

where $\alpha_i = F_i(0), \beta_i = H_i(0)$. For each of the other $AA_j$ with identity $aid_j$, $AA_i$ computes the subpartition of the master secret $s_{ij,1} = F_i(aid_j), s_{ij,2} = H_i(aid_j)$, and sends $(s_{ij,1}, s_{ij,2})$ to $AA_j$. $AA_i$ receives $n_\omega - 1$ subsegments $(s_{ji,1}, s_{ji,2})$ and then computes the main segmentation of the main secret.

$$s_{i,1} = sk_{\alpha_\omega,i} = \sum_{i=1}^{n_\omega} s_{ji,1}, \quad s_{i,2} = sk_{\alpha_\omega,i} = \sum_{i=1}^{n_\omega} s_{ji,2} \tag{2}$$

The segmentation of the public attribute key $pk_{i,\omega} = (e(g,g)^{sk_{\alpha_\omega,i}}, g^{sk_{\alpha_\omega,i}})$. Then $AA_i$ sends the $pk_{i,\omega}$ to the contract.

*Public Attribute Key Generation Contract*(*PAKGC*). AAs collaborate using the Shamir secret sharing scheme to generate the attribute public key for each attribute and upload it to the blockchain. Since multiple AAs jointly manage each attribute, the system needs a contract to collect the public attribute key slice generated by AAs. As shown in Algorithm 2, after the system initialization phase, the PAKGC collects $t_\omega$ public attribute key segmentation $pk_{i,\omega}$ from AAs and automatically invokes Lagrangian interpolation to calculate the public attribute key $PAK_\omega$ for encryption of data.

---

**Algorithm 2** Public attribute key generation

---

1: //AAs invoke SSS to generate the segmentation of public attribute key
2: **if** *segPAKGen*($AA_i$, $pk_{i,\omega}$) = true **then**
3:     $AA_i$ send $pk_{i,\omega}$ to Contract;
4:     Count[$\omega$]++;
5:     **if** Count[$\omega$] = t$_\omega$ **then**
6:         //This contract invokes Lagrangian interpolation method to generate $PAK_\omega$
7:         $PAK_\omega \leftarrow$ *Lagrangian*($t_\omega$, $pk_{i,\omega}$);
8:     **end if**
9: **end if**

---

$\boldsymbol{PAKGen(pk_i)} \rightarrow \boldsymbol{PAK}$: After receiving $t_\omega$ public attribute key splits $pk_{i,\omega}$, the PAKGC calculates

$$e(g,g)^{\alpha_\omega} = \prod_{i=1}^{t_\omega} e(g,g)^{sk_{\alpha_\omega,i} \prod_{j=1,j\neq i}^{t_\omega} \frac{aid_j}{aid_j - aid_i}}$$

$$g^{\beta_\omega} = g^{sk_{\beta_\omega,i} \prod_{j=1,j\neq i}^{t_\omega} \frac{aid_j}{aid_j - aid_i}}$$

(3)

The public attribute key $PAK_\omega = (e(g,g)^{\alpha_\omega}, g^{\beta_\omega})$, and the secret attribute key $SAK_\omega = (\alpha_\omega, \beta_\omega)$.

$\boldsymbol{DUSetup(GP)} \rightarrow \boldsymbol{(upk, usk)}$: Each data user who joins the system randomly chooses $usk \in Z_p^*$ to computes $upk = g^{usk}$. The user needs to upload $upk$ to the blockchain to obtain attribute authorization.

### 5.2. Decryption Key Generation Phase

As shown in Figure 4, AAs collaborate using the Shamir secret sharing scheme to generate the user attribute key segmentation and user attribute confused key segmentation off-chain. The UAKGC and the UACTGC reconstruct the user attribute key and user attribute confused key on-chain.

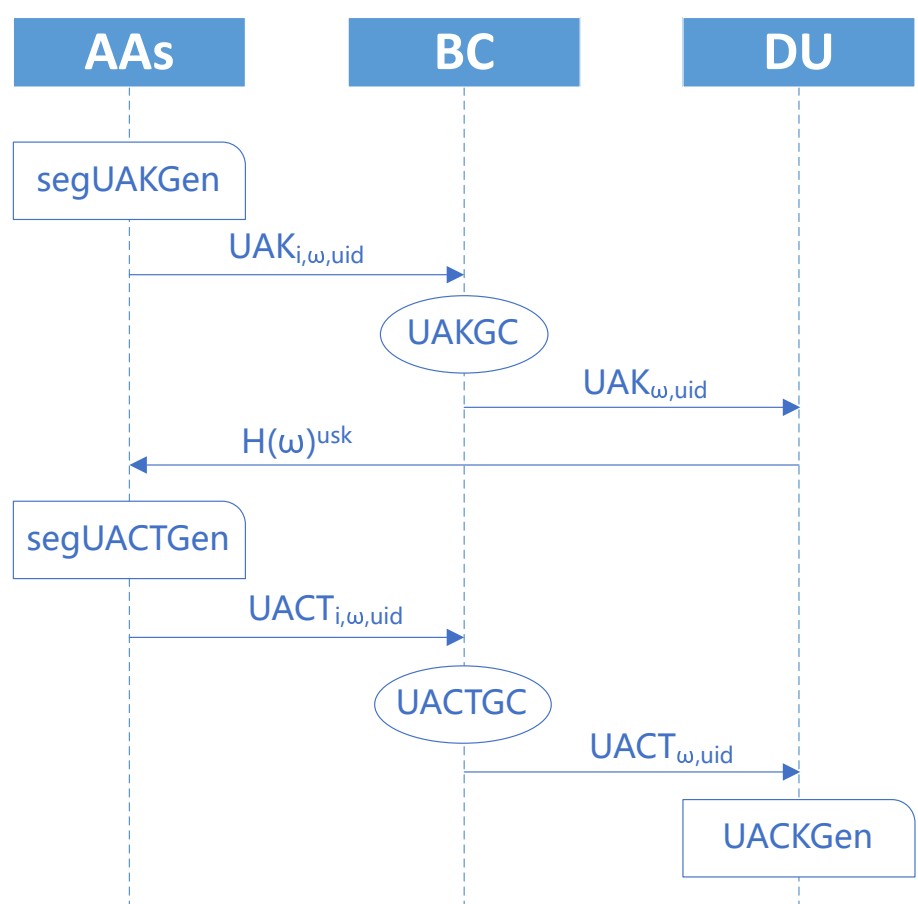

**Figure 4.** Decryption key generation phase.

*User Attribute Key Generation Contract (UAKGC).* When DU joins the system, AAs collaborate using the Shamir secret sharing scheme to generate the segmentation $UAK_{i,\omega,uid}$ of the user attribute key in their respective management domains. As shown in Algorithm 3, after the UAKGC collects $t_\omega$ $UAK_{i,\omega,uid}$ partitions from AAs, it automatically invokes the Lagrangian interpolation method to calculate the user attribute key $UAK_{\omega,uid}$ and then sends it to the user for data decryption.

---

**Algorithm 3** User attribute key generation

---

1: //AAs invoke SSS to generate the segment of user attribute key
2: **if** $segUAKGen(AA_i, UAK_{i,\omega,uid})$ = true **then**
3:      $AA_i$ send $UAK_{i,\omega,uid}$ to Contract;
4:      Count[$\omega$]++;
5:      **if** C **then**ount[$\omega$] = t$_\omega$
6:          //This contract invokes Lagrangian interpolation method to generate UAK$_\omega$
7:          $UAK_\omega \leftarrow Lagrangian(t_\omega, UAK_{i,\omega})$;
8:      **end if**
9: **end if**

---

$UAKGen(GP, S_{uid}) \rightarrow UAK$: This algorithm is classified into the following two phases.

(1)    When DU completes registration, $AA_i$ calculates the attribute key segmentation $UAK_{i,\omega,uid} = g^{sk_{\alpha\omega,i}} H(uid)^{sk_{\alpha\omega,i}}$, and sends the segmentation to the UAKGC.

(2)    After the UAKGC receives at least $t_\omega$ $UAK_{i,\omega,uid}$, we calculate

$$g^{\alpha_\omega} H(uid)^{\alpha_\omega} = \prod_{i=1}^{t_\omega} UAK_{i,\omega,uid}^{\prod_{j=1,j\neq i}^{t_\omega} \frac{aid_j}{aid_j - aid_i}} \tag{4}$$

The user attribute key $UAK_{\omega,uid} = g^{\alpha_\omega} H(uid)^{\alpha_\omega}$.

*User Attribute Confused Token Generation Contract (UACTGC).* The user computes $H(\omega)^{usk}$ for each attribute and sends it to each AA through a secure channel. As shown in Algorithm 4, AAs generate the segmentation $UACT_{i,\omega,uid}$ of the user attribute confused token in their respective administrative domains. After the UACTGC receives at least $t_\omega$ $UACT_{i,\omega,uid}$ from the AAs, it automatically invokes Lagrangian interpolation to compute the user attribute confused token $UACT_{\omega,uid}$.

---

**Algorithm 4** User attribute confused token generation

---

1: //AAs invoke SSS to generate the segment of user attribute confused token
2: **if** $segUACTGen(AA_i, UACT_{i,\omega,uid})$ = true **then**
3:      $AA_i$ send $UACT_{i,\omega,uid}$ to Contract;
4:      Count[$\omega$]++;
5:      **if** C **then**ount[$\omega$] = t$_\omega$;
6:          //This contract invokes Lagrangian interpolation method to generate UACT$_\omega$
7:          $UACT_\omega \leftarrow Lagrangian(t_\omega, UACT_{i,\omega})$;
8:      **end if**
9: **end if**

---

$UACKGen(GP, S_{uid}) \rightarrow UACK$: This algorithm is classified into the following three phases.

(1)    DU computes $H(\omega)^{usk} \forall \omega \in [S_{uid}]$, where $S_{uid}$ is the attribute in the user attribute set, and then sends $H(\omega)^{usk}$ to each AA through a secure channel. To verify DU's ownership of the attribute, AAs confirm $e(H(\omega), upk) = e(H(\omega)^{usk}, g)$. After successful verification, AAs share the secret between each other secret sharing, compute the partition $UACT_{i,\omega,uid} = H(\omega)^{usk\cdot\delta_i}$ of the confused key token for attribute $\omega$, and send this partition to the UACTGC.

(2)    After the UACTGC collects at least $t_\omega UACT_{i,\omega,uid}$, it computes

$$H(\omega)^{usk\cdot\delta} = \prod_{i=1}^{t_\omega} H(\omega)^{usk\cdot\delta_i \prod_{j=1,j\neq i}^{t_\omega} \frac{aid_j}{aid_j - aid_i}} \tag{5}$$

The user attribute confused key $UACT_{\omega,uid} = H(\omega)^{usk\cdot\delta}$.

(3) The user decrypts with his private key to get the attribute confused key $SK_C = UACK_{\omega,uid} = [UACT_{\omega,uid}]^{1/usk} = H(\omega)^{\delta}$.

### 5.3. Encryption Phase

As shown in Figure 5, the DO is responsible for encrypting the medical data.

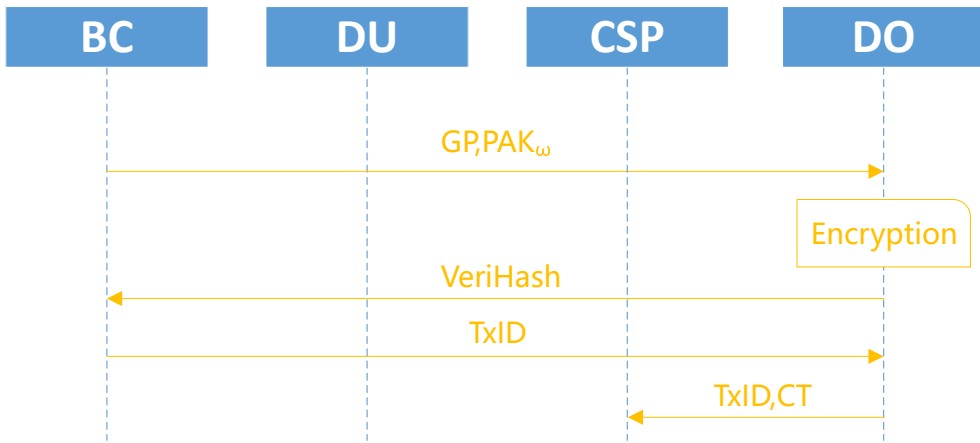

**Figure 5.** Encryption phase.

$\textbf{\textit{Encryption}}(\textbf{\textit{KEY}}, \textbf{\textit{M}}_{\textbf{\textit{data}}}, \textbf{\textit{AP}}, \textbf{\textit{GP}}, \textbf{\textit{PAK}}) \rightarrow \textbf{\textit{CT}}$: The data user first encrypts plaintext with a symmetric key and then encrypts the symmetric key with *the linear secret sharing scheme (LSSS)*. First, DO generates a symmetric key $KEY \in G_2$ to encrypt the data $AESEncrypt(KEY, M_{data}) = CT_{AES}$. Then DO defines an access policy $AP$ and converts $AP$ into an *LSSS* shared generation matrix over the associated attributes and a mapping function: $(M_{l \times m}, \rho)$, where $M$ is an $l \times m$ matrix. The function $\rho$ maps the $x$th row of $M$ to the attributes $\rho(x) \in \{attr_1, \ldots, attr_j\}$, where $j \in [J]$, $J$ is an attribute in the access policy $AP$, and $attr_j$ is an attribute in the access policy. DO generates the ciphertext according to the following two steps.

(1) To hide the attributes in the matrix and protect the privacy of the data sharing parties, DO needs to confuse the attribute mapping function: DO randomly chooses $r \in Z_p^*$, computes $\sigma_j = e((g^{\delta})^r, H(attr_j)) \forall j \in [J]$. DO substitutes $\sigma_j$ for the attributes mapped by the rows of the shared generation matrix: $\rho'(x) = e((g^{\delta})^r, H(attr_x))$, where $x \in l$. The access policy $AP$ is then transformed into an LSSS scheme $\Pi : (M_{l \rightarrow m}, \rho')$ over the attributes of interest.

(2) For each row of the matrix $M$, DO randomly selects a set of vectors $v = (s, r_1, r_2, \ldots, r_m)$, where the secret values $s \in Z_p^*$, and $r_1, r_2, \ldots, r_m \in Zp*$. Let $\lambda_x = M_x v^T$ where $M_x$ is the $x$th row of the matrix M. DO randomly select a set of vectors $h = (0, h_1, h_2, \ldots, h_m) \in Z_p^*$, compute $c_x = M_x h^T$. And randomly choosing $u_1, u_2, \ldots, u_l \in Z_p^*$. Then the ciphertext is calculated as
$CT_{KEY} = KEY \cdot e(g, g)^{y \cdot s}$
$C_{1,x} = e(g, g)^{\alpha_{\rho(x)} \cdot u_x}$
$C_{2,x} = g^{u_x}$
$C_{3,x} = g^{c_x} \cdot g^{\beta_{\rho(x)} \cdot u_x}$
$C_{4,x} = g^{c_x} \cdot g^{y \cdot \lambda_x}$
$C_v = H(e(g, g)^{y \cdot s})$

where $C_v$ is used to verify the correctness of the outsourced decryption result. DO uploads the verification hash $C_v$ to the blockchain, and then the blockchain returns the transaction identifier TxID. Finally, DO uploads the ciphertext $CT = ((M, \rho'), g^r, CT_{KEY}, C_{1,x}, C_{2,x}, C_{3,x}, C_{4,x}, C_v)$ and TxID to the cloud CSP.

### 5.4. Decryption Phase

As shown in Figure 6, CSP is responsible for the outsourced data decryption, and DO verifies the outsourced decryption results and performs the final decryption.

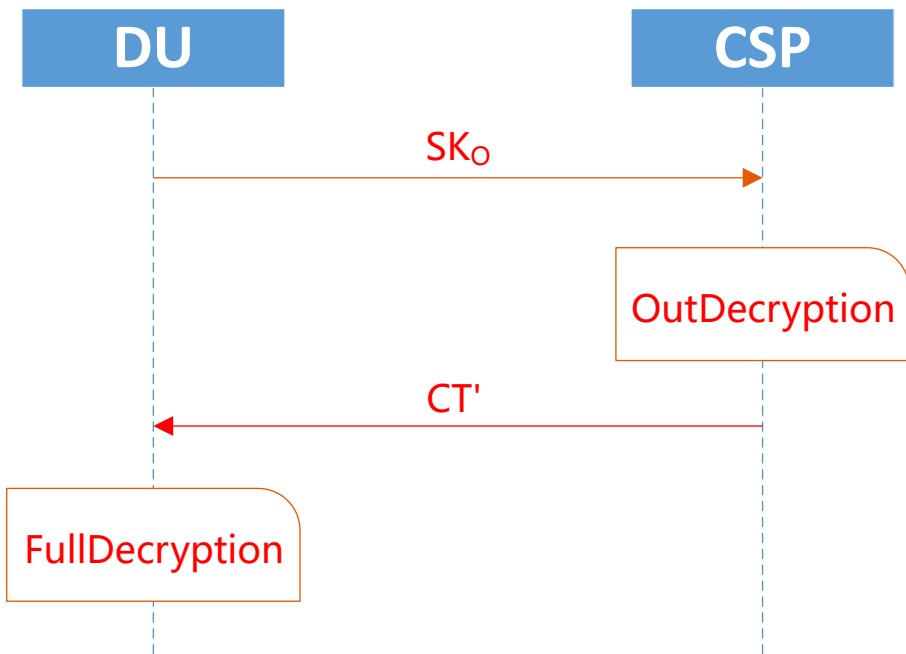

**Figure 6.** Decryption phase.

$OutDecryption(CT, SK_O, AP) \rightarrow CT'$: This algorithm is classified into the following two phases.

(1) To view the data of DO, DU initiates an access request to the blockchain and then downloads the $CT$ from the CSP. DU calculates $\{\sigma_\omega : \sigma_\omega = e((g)^r, H(attr_\omega)^\delta)\}_{\forall\omega\in[S_{uid}]}$, where $S_{uid}$ is the number of attributes in the user attribute set, based on $(M, \rho')$ and $g^r$ in the ciphertext and $UACK_{uid}$. DU computes the line number $X : \{\sigma_\omega\}_{\forall\omega\in[S_{uid}]} \cap \{\rho'(x)\}_{\forall x\in[l]}$, and sends the outsourced decryption key $SK_O = UAK_{\rho'(x),uid}$ to the CSP, where $x \in X$.

(2) If the user satisfies the access policy, then $M_x$ must be a full rank matrix. CSP finds a set $t_x \in Z_p^*$ in polynomial time such that $\sum_{x\in X} t_x M_x = (1, 0, \ldots, 0)$, where $M_x$ corresponds to the set of attributes satisfying the policy $S_x$, and $t_x$ helps to recover the secret value $s$. The CSP then performs outsourced decryption based on the received user attribute key.

$$CT' = \prod_x (C_{1,x} \cdot \frac{e(H(uid), C_{3,x})}{e(SK_O, C_{2,x})} \cdot e(upk, C_{4,x}))^{t_x}$$
$$= e(g, g)^{usk \cdot y \cdot s} \tag{6}$$

Then, the CSP returns the outsourced decryption result $CT'$ to DU.

$FullDecryption(CT', usk) \rightarrow M_{data}$: To verify the correctness of the outsourced decryption result, DU performs an exponential operation with its private key to get $C'_v = e(g, g)^{y \cdot s}$ and then calculates $H'(C'_v)$. Compared with the $C_v$ in the ciphertext, if $H'(C'_v) = C_v$, the outsourced decryption result is correct. DU continues to compute the symmetric key $KEY = CT_{KEY}/C'_v$ and then performs AES decryption on the data: $AESDecrypt(KEY, CT_{AES}) \rightarrow M_{data}$.

## 6. Scheme Analysis

*6.1. Correctness Analysis*

6.1.1. The Correctness of CSP Outsourcing Decryption

After the CSP receives the outsourced decryption key from DU, it calculates

$$
\begin{aligned}
CT' &= \prod_x (C_{1,x} \cdot \frac{e(H(uid), C_{3,x})}{e(SK_O, C_{2,x})} \cdot e(upk, C_{4,x}))^{t_x} \\
&= \prod_x (e(g,g)^{\alpha_{\rho'(x)} \cdot u_x} \cdot \frac{e(H(uid), g^{c_x} \cdot g^{\beta_{\rho'(x)} \cdot u_x})}{e(UAK_{\rho'(x),uid}, g^{u_x})} \cdot e(g^{usk}, g^{c_x} \cdot g^{y \cdot \lambda_x}))^{t_x} \\
&= \prod_x (\frac{e(g,g)^{\alpha_{\rho'(x)} \cdot u_x} \cdot e(H(uid), g^{c_x}) \cdot e(H(uid), g^{\beta_{\rho'(x)} \cdot u_x}) \cdot e(g^{usk}, g^{c_x}) \cdot e(g^{usk}, g^{y \cdot \lambda_x})}{e(g^{\alpha_{\rho'(x)}}, g^{u_x}) \cdot e(H(uid)^{\beta_{\rho'(x)}}, g^{u_x})})^{t_x} \\
&= \prod_x (e(g,g)^{usk \cdot y \cdot \lambda_x} \cdot e(H(uid), g)^{c_x} \cdot e(g,g)^{usk \cdot c_x})^{t_x}
\end{aligned}
\tag{7}
$$

Since $\lambda_x = M_x v^T, c_x = M_x h^T, v^T = (s, r_1, r_2, \ldots, r_m)^T, h^T = (0, h_1, h_2, \ldots, h_m)^T$

$$
\begin{aligned}
\lambda_x \cdot t_x &= M_x t_x \cdot v^T = (1, 0, \ldots, 0) \cdot (s, r_1, r_2, \ldots, r_m)^T = s; \\
c_x \cdot t_x &= M_x t_x \cdot h^T = (1, 0, \ldots, 0) \cdot (0, h_1, h_2, \ldots, h_m)^T = 0
\end{aligned}
\tag{8}
$$

Therefore, the outsourced decryption ciphertext $CT' = e(g,g)^{usk \cdot y \cdot s}$, and the CSP outsourcing decryption algorithm satisfies the correctness.

6.1.2. The Correctness of DU Decryption

After receiving the outsourced decryption cipher, DU calculates

$$
\begin{aligned}
KEY &= \frac{CT_{KEY}}{[CT']^{1/usk}} \\
&= \frac{KEY \cdot e(g,g)^{y \cdot s}}{[e(g,g)^{usk \cdot y \cdot s}]^{1/usk}} \\
&= \frac{KEY \cdot e(g,g)^{y \cdot s}}{e(g,g)^{y \cdot s}}
\end{aligned}
\tag{9}
$$

Therefore, DU can obtain the symmetric key, and the DU decryption algorithm satisfies the correctness.

*6.2. Security Analysis*

6.2.1. Policy Hidden Security

The patient uploads the encrypted data to the cloud server and replaces the attribute $attr_j$ in the access policy matrix with an implicit bilinear mapping expression $\sigma_j = e((g^\delta)^r, H(attr_j))$, which is embedded in the ciphertext using a one-way hash function. Only an authorized user with the key $H(\omega)^\delta$ can compute $e(g^r, H(\omega)^\delta)$. Since $\delta$ is a random value, the cloud server and other users cannot guess the attribute $\omega$ from the value $e(g^r, H(\omega)^\beta)$. Without knowing the corresponding $H(\omega)^\delta$, no one can compute $e((g^\delta)^r, H(\omega)) = e(g^r, H(\omega)^\delta)$ and hence cannot construct the attribute in the access policy.

6.2.2. System Robustness

We use the Shamir secret sharing scheme $(t, n)$ to manage user attributes and distribute user attribute keys, so there is no single point of failure in the system. However, the adversary may bring the system down by attacking multiple AAs. If there are fewer than t AAs in the system, the system will go down. The system's robustness is affected by the threshold $t$ and the total number $n$. A threshold $t$ that is too large introduces additional overhead. Considering the system's security, we can set a suitable threshold value to make the system resist the attack with a high probability.

### 6.2.3. Security against Collusion Attack

Multiple malicious users conspire with each other to share their own attribute keys $e(H(uid), g)^{c_x}$ in an attempt to decrypt other secrets. Suppose two users with identities $uid$ and $uid'$, respectively, try to conspire to merge their secrets $e(H(uid), g)^{c_x}$ and $e(H(uid), g)^{c'_x}$. They cannot cancel each other and thus cannot continue to recover out $e(g, g)^{usk \cdot y \cdot s}$.

### 6.2.4. Adaptive Selection of Ciphertext Indistinguishable Security under Chosen Plaintext Attack (INDS-CPA)

Let $G_1$ and $G_2$ be two multiplicative cyclic groups of order $p$ and satisfy the bilinear mapping $e : G_1 \times G_1 \to G_2$, $g$ is the $G_1$ generating elements, and randomly selected $a, b, s, z \in Z_p^*$. No probabilistic polynomial-time algorithm can distinguish the tuples $[g, g^a, g^b, g^s, e(g, g)^{abs}]$ and the tuples $[g, g^a, g^b, g^s, e(g, g)^z]$ by a non-negligible advantage, called the DBDH assumption. If the DBDH problem is intractable, then the VO-PH-MAABE scheme satisfies the indistinguishable security under the chosen ciphertext attack. It includes adversary $\mathfrak{A}$ and challenger $\mathfrak{B}$, as follows.

***Initialization***: Adversary $\mathfrak{A}$ submits a challenge access policy $(M^*, \rho^*)$ to Challenger $\mathfrak{B}$.

***Setup***: Challenger $\mathfrak{B}$ chooses a sufficiently safe parameter $\lambda$ to run the $Setup(1^\lambda)$ algorithm to generate the public parameter $GP$. Challenger $\mathfrak{B}$ sends $GP$ to adversary $\mathfrak{A}$.

***Query Phase* 1**: The adversary adaptively queries the following Oracle.

$O_{AAs}$: Adversary $\mathfrak{A}$ specifies a set of corrupted permissions, and for each attribute $\omega$ belonging to the uncorrupted permissions, $\mathfrak{B}$ runs $AASetup(GP, \omega)$ to generate the public attribute key $(e(g, g)^{\alpha_\omega}, g^{\beta_\omega})$ and returns it to $\mathfrak{A}$.

$O_{UAK}$: Adversary $\mathfrak{A}$ submits an identity and its corresponding attribute set $(uid, S^*)$ to $\mathfrak{B}$. When the attribute set is insufficient for the challenge access strategy, adversary $\mathfrak{B}$ runs $UAKGen(uid, (M^*, \rho^*), S^*, GP)$ to generate the attribute key $UAK_{\omega, uid}$, and returns it to $\mathfrak{A}$.

***Challenge***: $\mathfrak{A}$ submits two equal-length symmetric keys $KEY_0$ or $KEY_1$ to $\mathfrak{B}$. $\mathfrak{B}$ rolls a coin to determine the value of $\gamma \in \{0, 1\}$ and executes $Encrypt(KEY_\gamma, M_{data}, (M^*, \rho^*), GP, e(g, g)^{\alpha_\omega}, g^{\beta_\omega})$. $\mathfrak{B}$ computes $CT_{KEY} = KEY \cdot e(g, g)^{abs}$, and for each row $M^*$ in the matrix $M_x^*$, $\mathfrak{B}$ randomly chooses $v = (s, r_1, r_2, \ldots, r_m) \in Z_p^*$, $h = (0, h_1, h_2, \ldots, h_m) \in Z_p^*$, where $r_1, r_2, \ldots, r_m, h_1, h_2, \ldots, h_m$ are chosen randomly from $Z_p^*$, and computes $\lambda_x = M_x v^T$ and $c_x = M_x h^T$. In addition, $\mathfrak{B}$ randomly chooses $z, u_1, u_2, \ldots, u_l \in Z_p^*$ and computes $C_0 = e(g, g)^z, C_{1,x} = e(g, g)^{\alpha_{\rho(x)} \cdot u_x}, C_{2,x} = g^{u_x}, C_{3,x} = g^{c_x} \cdot g^{\beta_{\rho(x)} \cdot u_x}, C_{4,x} = g^{c_x} \cdot g^{y \cdot \lambda_x}$. The challenge message $CT^* = ((M^*, \rho^*), g^r, CT_{KEY}, C_{1,x}, C_{2,x}, C_{3,x}, C_{4,x})$ and sends it to $\mathfrak{A}$.

***Query Phase* 2**: Adversary $\mathfrak{A}$ continues the adaptive query according to *Query Phase* 1.

***Guess***: Adversary $\mathfrak{A}$ makes a guess $\gamma' \in \{0, 1\}$ for $\gamma$. If $\gamma = \gamma'$, then challenger $\mathfrak{B}$ returns 1, and $Z = e(g, g)^{abs}$; if $\gamma \neq \gamma'$, then $\mathfrak{B}$ returns 0, and $Z = e(g, g)^z$. If $Z = e(g, g)^{abs}$, then $Pr[\gamma = \gamma'] = 1/2 + \varepsilon$; if $Z = e(g, g)^z$, then $Pr[\gamma = \gamma'] = 1/2$. Thus, the probability that $\mathfrak{B}$ solves the problem is $Pr_{\mathfrak{B}, DBDH} = 1/2 Pr[\gamma = \gamma' | Z = e(g, g)^{abs}] - 1/2 Pr[\gamma = \gamma' | Z = e(g, g)^z] = \varepsilon/2$.

In polynomial time, the advantage of challenger $\mathfrak{B}$ in breaking the DBDH puzzle is negligible, and similarly, the advantage of $\mathfrak{A}$ in winning the game $\epsilon$ is also negligible. Thus, this scheme (VO-PH-MAABE) is INDS-CPA safe.

### 6.3. Performance Analysis

This section compares and analyzes the existing scheme with this paper's scheme in terms of pass-through feature, communication overhead, and computation overhead. The experiments of this scheme are conducted on Windows 11 with Intel(R) Core(TM) i5-10505CPU @ 3.20 GHz and 16GB RAM. We use Java18 developed by Oracle Corporation of America and a 256-bit elliptic curve-based JPBC cryptography library to simulate this scheme.

6.3.1. Feature Comparison

As shown in Table 1, we compare our scheme with schemes [6,7,21,24] and [26] in terms of how attribute authority works, underlying blockchain architecture, policy hiding, outsourcing decryption, outsourcing verification, and smart contracts. In terms of attribute authority, schemes [21,26] have assumed trusted CAs and suffer from concentration of power. Although there is no centralization problem in scheme [7], each AA in the scheme manages the attributes in the domain independently of each other, so the failure of any AA affects the attribute management. The attribute Universe in other schemes is managed by a group of AAs through joint negotiation, which can effectively avoid a single point of failure and enhance system security. Scheme [24] and our scheme combine blockchain systems. In terms of protecting sensitive attributes, only scheme [6], scheme [7], and our scheme implement policy hiding to protect users' privacy. In terms of outsourcing, schemes [21,24,26] and our scheme implement outsourced decryption, outsourcing the complex decryption operations to cloud servers to reduce the computational overhead at the user side. However, schemes [24,26] are unable to verify the correctness of the results after outsourcing decryption. Besides, only this scheme achieves distributed management of attributes and cross-domain computation of keys by using smart contracts.

**Table 1.** Feature Comparison.

| Scheme | Attribute Authority | Blockchain | Policy Hiding | Outsourcing Decryption | Outsourcing Verification | Smart Contracts |
|---|---|---|---|---|---|---|
| Qin [21] | Single | × | × | √ | √ | × |
| TMACS [6] | Multi-Common | × | √ | × | × | × |
| dec-ABE [7] | Multi-Common | √ | × | √ | × | × |
| BC-SABE [24] | Multi-Exclusive | × | √ | × | × | × |
| O3-R-CP-ABE [26] | Single | × | × | √ | × | × |
| Our scheme | Multi-Common | √ | √ | √ | √ | √ |

6.3.2. Communication Overhead

Let $|G_1|$ and $|G_2|$ be the size of the elements in the multiplicative cyclic groups $G_1$, $G_2$, $|Z_p^*|$ denote the size of the elements in the domain $Z_p^*$, $N_{attr}$ denotes the number of attributes contained in the parameters, $N_u$ denotes the number of user attributes, $U$ denotes the number of attributes in the attribute space, $L_v$ denotes the size of the authentication parameters, and $N$ and $n$ denote the number of representative users and AAs, respectively. As shown in Table 2, we compare our scheme with existing schemes in terms of public parameters $GP$, size of DU complete attribute key $SK$, size of ciphertext $CT$, and size of outsourced decryption ciphertext $CT'$. During the system initialization phase, our scheme generates a public parameter $GP$ with a size of $3|G_1|+|G_2|$, which is constant compared to the scheme [21] and scheme [6]. In our scheme, the complete attribute private key of DU is $SK = \{SK_C, SK_O, usk\}$, so the length of $SK$ is $2|G_1|+|Z_p^*|$, which is constant compared to schemes [6,21,24,26]. The size of the ciphertext $CT$ in our scheme is $(3N_{attr} + 1)|G_1| + (N_{attr} + 1)|G_2|$, and our scheme hides the access policy, so it has a slightly higher communication burden relative to the other parties. Since the encryption algorithm is only a one-time operation, our scheme does not affect the user experience. Moreover, the size of the outsourced decryption ciphertext in our scheme is $|G_2|$, and the size of the outsourced decryption ciphertext sent to DU is shorter compared to scheme [21].

**Table 2.** Communication overhead.

| Scheme | GP | SK | CT | CT' |
|---|---|---|---|---|
| Qin [21] | $(2+U)\|G_1\|+\|G_2\|$ | $2\|G_1\|+N_u\|G_2\|$ | $(2Nattr+1)\|G_1\|+\|G_2\|+L_v$ | $2\|G_2\|$ |
| TMACS [6] | $(2U+2N+n+10)\|Z_p^*\|$ | $(2N_u+5)\|Z_p^*\|$ | $(2N_{attr}+1)\|G_1\|+\|G_2\|$ | $-$ |
| dec-ABE [7] | $2\|G_1\|+\|G_2\|$ | $(N_u+3)\|G_1\|$ | $(2N_{attr}+2)\|G_1\|+(N_{attr}+1)\|G_2\|$ | $\|G_2\|$ |
| BC-SABE [24] | $12\|G_1\|+\|G_2\|+\|Z_p^*\|$ | $\|Z_p^*\|$ | $(3N_{attr}+2)\|G_1\|+\|G_2\|$ | $-$ |
| O3-R-CP-ABE [26] | $3\|G_1\|+\|G_2\|$ | $(N_u+2)\|G_1\|+\|G_2\|$ | $2N_{attr}\|G_1\|+\|G_2\|$ | $\|G_2\|$ |
| Our scheme | $3\|G_1\|+\|G_2\|$ | $2\|G_1\|+\|Z_p^*\|$ | $(3N_{attr}+1)\|G_1\|+(N_{attr}+1)\|G_2\|$ | $\|G_2\|$ |

### 6.3.3. Computation Overhead

As shown in Figure 7, we compare our scheme with schemes [6,7] and scheme [24] in terms of the computational overhead of decryption on the user side. The horizontal axis is the number of attributes for which the DU satisfies the access policy, and the vertical axis is the time consumed for decryption. Assume that before decryption, the DU has calculated the attributes needed for decryption by the attribute confused key and $g^r$. For the same $g^r$, the overhead in the subsequent decryption process is negligible since the process only needs to be calculated once permanently. During the decryption process, the DU only needs to perform a power operation to recover the symmetric key because the cloud server performs the outsourced decryption process. With the number of attributes increasing, compared with scheme [6,7] in which the decryption time increases linearly with the increase of attribute number, our scheme revolves around 0.043 s. So, our scheme has higher decryption efficiency compared with schemes [6,7].

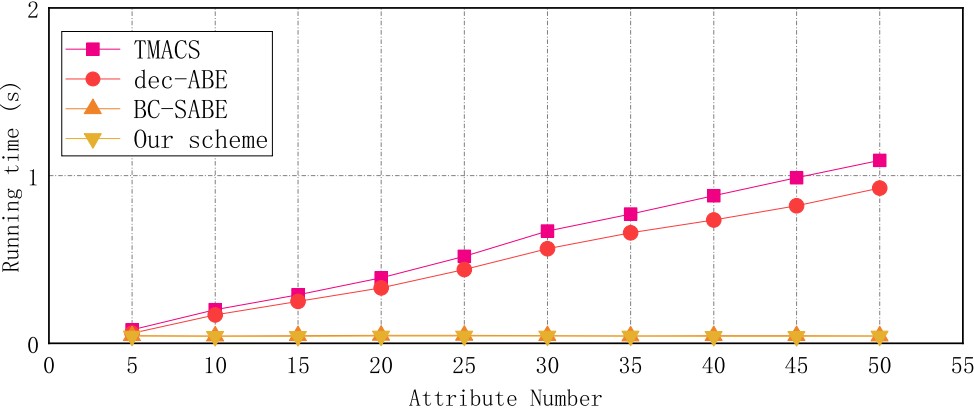

**Figure 7.** The comparison of our scheme with the TMACS scheme [6], dec-ABE scheme [7], and BC-SABE scheme [24] in terms of user decryption time.The comparison of user decryption time.

### 6.3.4. Blockchain Simulation Experiments

To verify the effectiveness of smart contracts, we test the performance of smart contracts using the private chain simulation of Ethereum.

In this experiment, we set the number of authorities to five. As shown in Figure 8, the horizontal axis is the attributes managed by AAs and the vertical axis is the response latency of the smart contract. The AMC does not have any computation and only involves the assignment of attributes, so it does not grow linearly with the number of attributes but always remains around 0.45 s. The other three contracts run a Lagrangian interpolation algorithm each time the key is reconstructed, so the response latency of the PAKGC, the UAKGC, and the UACTGC grows slowly with the number of attributes. The other three contracts run a Lagrangian interpolation algorithm each time the key is reconstructed, so the response latency of the PAKGC, the UAKGC, and the UACTGC grows slowly with the number of attributes. At the number of attributes of 50, the response latencies of PAKGC, UAKGC, and UACTGC are 1.818 s, 1.702 s, and 1.659 s, respectively. Since it does not affect the user experience, it is acceptable.

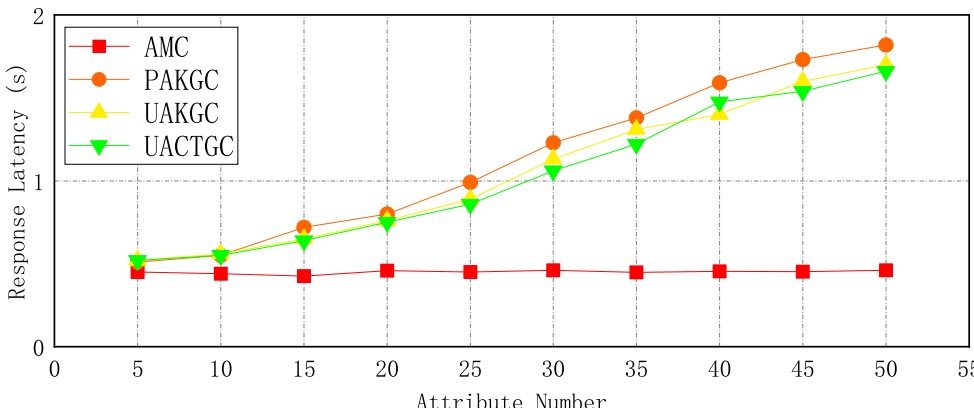

**Figure 8.** Response latency of the four contracts in our scheme.

## 7. Conclusions

In this paper, we proposed a blockchain-based multiple authority attribute-based encryption EHR access control scheme to solve the access control and security problem of data in the IOMT environment. Using blockchain smart contracts, we have achieved cross-domain distributed management of attributes and cross-domain computation of different authorities, which reduces the cost of cross-domain computation and eliminates the single-point bottleneck problem of traditional CP-ABE schemes. In the encryption phase, DO hid the access policy to ensure that the privacy of data users is not leaked to third parties. In the decryption phase, the computational cost on the user side is reduced by outsourcing the decryption algorithm. Under the random oracle model, the ciphertext indistinguishability security under the adaptive selection of plaintext attacks has been proven. In addition, the scheme algorithm's efficiency has been analyzed using the JPBC cryptography library, and the feasibility of smart contracts has been demonstrated using blockchain simulation experiments.

**Author Contributions:** All authors contributed to this work. Writing—review and editing, X.Y.; writing—original draft preparation, C.Z. All authors have read and agreed to the published version of the manuscript.

**Funding:** This work was supported by Ministry of Science and Technology of China, National Key R&D Program "Cyberspace Security" Key Project, 2017YFB0802305 and the Natural Science Foundation of Hebei Province, F2021201052.

**Institutional Review Board Statement:** Not applicable.

**Informed Consent Statement:** Not applicable.

**Data Availability Statement:** Not applicable.

**Conflicts of Interest:** The authors declare no conflict of interest.

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
