# Peer review of "Blockchain-Based Multiple Authorities Attribute-Based Encryption for EHR Access Control Scheme"

_applsci, doi:10.3390/app122110812_

Round 1

Reviewer 1 Report

The paper presents a blockchain-based multiple authorities attribute-based encryption scheme for EHR access control. The authors combined an attribute-based encryption along with verifiable outsourcing decryption and hiding access policies for effective privacy-preserving of EHR. The overall paper is well-written and there are minor concerns to be highlighted:

1-    Captions in the performance analysis section should have more explanation in the figures.

2-    English proofreading is recommended and the quality of figures should be improved.

Reviewer 2 Report

1. Need to modify some of the acronyms that are not primarily introduced before being used in the paragraph for example CP-ABE, KP-ABE, CP-AB etc.

2. In section 5, replace 'chapter' with 'section'

3. In section 5, change the sub-title 'specific scheme' to 'scheme implementation'.

4. Need to add the significant value that you gained (how many seconds?) in the experiment done in sections 6.3.2, 6.3.3, and 6.3.4 rather than stating the general statement of the achievement. This value can be added to the abstract to show your significant contribution to this research.
